# Application research of artificial intelligence software in the analysis of thyroid nodule ultrasound image characteristics

**Chen Xu[1]☯, Zuxin Wang[1]☯, Jun Zhou[2], Fan Hu[1], Ying Wang[1], Zhongqing Xu[3]\*, Yong Cai[1,4]\***

**1** Public Health Research Center, Tongren Hospital, Shanghai Jiao Tong University School of Medicine, Shanghai, P.R. China, **2** Project Department, Tend.AI Medical Technologies Co., Shanghai, P.R. China, **3** Department of General Practice, Tongren Hospital, Shanghai Jiao Tong University School of Medicine, Shanghai, P.R. China, **4** Center for Community Health Care, China Hospital Development Institute, Shanghai Jiao Tong University, Shanghai, China

☯ These authors contributed equally to this work and share first authorship.
\* zhongqing_xu@126.com (ZX), caiyong202028@hotmail.com (YC)

## Abstract

Thyroid nodule, as a common clinical endocrine disease, has become increasingly prevalent worldwide. Ultrasound, as the premier method of thyroid imaging, plays an important role in accurately diagnosing and managing thyroid nodules. However, there is a high degree of inter- and intra-observer variability in image interpretation due to the different knowledge and experience of sonographers who have huge ultrasound examination tasks everyday. Artificial intelligence based on computer-aided diagnosis technology maybe improve the accuracy and time efficiency of thyroid nodules diagnosis. This study introduced an artificial intelligence software called SW-TH01/II to evaluate ultrasound image characteristics of thyroid nodules including echogenicity, shape, border, margin, and calcification. We included 225 ultrasound images from two hospitals in Shanghai, respectively. The sonographers and software performed characteristics analysis on the same group of images. We analyzed the consistency of the two results and used the sonographers' results as the gold standard to evaluate the accuracy of SW-TH01/II. A total of 449 images were included in the statistical analysis. For the seven indicators, the proportions of agreement between SW-TH01/II and sonographers' analysis results were all greater than 0.8. For the echogenicity (with very hypoechoic), aspect ratio and margin, the kappa coefficient between the two methods were above 0.75 (P < 0.001). The kappa coefficients of echogenicity (echotexture and echogenicity level), border and calcification between the two methods were above 0.6 (P < 0.001). The median time it takes for software and sonographers to interpret an image were 3 (2, 3) seconds and 26.5 (21.17, 34.33) seconds, respectively, and the difference were statistically significant (z = -18.36, P < 0.001). SW-TH01/II has a high degree of accuracy and great time efficiency benefits in judging the characteristics of thyroid nodule. It can provide more

**Data availability statement:** The data set used for analysis in this study has been uploaded as Supporting Information files.

**Funding:** This study was supported by the Clinical Research Project of Shanghai Municipal Health Commission (202240198), Shanghai Municipal Health Commission Clinical

Research Project (202140203), Key Discipline Projects of Shanghai Three-Year Action Plan for Public Health under Grant (GWVI-11.1-29), Science and Technology Commission Shanghai Municipality (Grant 20JC1410204) for the Seroepidemiological Study of Novel Coronavirus Pneumonia in Key Populations, Key Supporting Disciplines of Shanghai Health System (Grant Number-2023ZDFC0403), Shanghai Health Care Commission Clinical Research Program (20214Y0205). The funders contributed to the study design, data collection and analysis, preparation of the manuscript, and the decision to publish.

**Competing interests:** The authors have declared that no competing interests exist.

objective results and improve the efficiency of ultrasound examination. SW-TH01/II can be used to assist the sonographers in characterizing the thyroid nodule ultrasound images.

## Introduction

Thyroid nodule (TN), as a common clinical endocrine disease, has become increasingly prevalent worldwide with a detection rate of 19%-68% [1]. Notably, approximately 7%-15% of patients with TNs can develop thyroid cancer (TC) [2]. Epidemiological surveys highlight a rising incidence of TC, making it one of the fastest-growing malignancies [3]. Early diagnosis of TN is clinically significant with the need to exclude thyroid cancer. However, most TNs are easily overlooked in the early stage due to the absence of typical symptoms and signs, risking the possibility of missing the optimal treatment window. A reliable method is needed to accurately differentiate between malignant and benign nodules, to help clinicians identify which TNs require further attention or intervention.

With the rapid advancement of imaging devices and technologies, ultrasound has emerged as the premier method of imaging for the thyroid, playing an important role in accurately diagnosing and managing TNs [1]. According to the American College of Radiology (ACR) Thyroid Imaging Reporting and Data System (TI-RADS), five key grey-scale features are used to assess the risk category of TNs, including nodule consistency, echogenicity, shape, margins, and the presence or absence of echogenic foci or calcifications [4]. Solid hypoechoic or partially cystic hypoechoic nodules with microcalcifications, irregular margins, extrathyroidal extension, taller-than-wide shape, or rim calcifications with extrusive soft tissue component were designated high suspicion nodules by the 2015 American Thyroid Association (ATA) guidelines [5]. Individual features are not enough to diagnose malignant changes, but the sensitivity of diagnosing malignant changes increases if two or more of these features are present simultaneously [6,7]. Therefore, a comprehensive analysis of the main ultrasound features of TNs is of great clinical importance and value in diagnosing TNs.

Despite the advantages of ultrasound being non-invasive, convenient, and inexpensive, there are still some problems in clinical practice. Firstly, the analysis of medical images via artificial means involves a massive workload, and accuracy can easily be impacted by subjective factors [8]. Additionally, standardized quantification criteria are often absent, leading to misdiagnosis, missed diagnosis, or overtreatment of TNs. Research has shown that there is a global trend of overdiagnosis of papillary thyroid carcinoma, with overdiagnosis rates as high as 93% in South Korea and approximately 87% in China [9]. Secondly, there is a shortage of medical professionals in medical imaging. The annual growth rate of medical image data in China is about 30%, while the rate of sonographers is only around 4% [10]. Finally, the uneven distribution of medical resources and significant differences in medical levels across different regions in China means that some ultrasound doctors may lack experience and technical skills, leading to misjudgment.

Artificial Intelligence (AI) has made significant strides in the field of medicine due to the development of computer technology, electronic engineering, statistics, and other disciplines [11]. Through the analysis of vast datasets of medical images, AI algorithms can recognize highly discriminatory image features, thereby enabling them to analyze target lesions accurately [12]. In recent years, due to the availability of large datasets and the urgency of clinical demands, AI technology has developed rapidly in diagnosing TNs. Wang et al. used a neural network to build an automatic image recognition and diagnosis system, and the results showed that compared with the performance of experienced sonographers, the AI system has comparable sensitivity (90.50%) and accuracy (90.31%) for the diagnosis of TNs, and higher specificity [13]. Peng's paper showcases the feasibility of incorporating AI into managing TNs using ThyNet. The application of ThyNet significantly enhanced sonographers' diagnostic accuracy with pooled AUC improving from 0.837 to 0.875 [14].

TenD AI Medical Technology (Shanghai) Co., Ltd. developed an AI software called SW-TH01/II to evaluate ultrasound image characteristics of TNs including echogenicity, shape, border, margin, and calcification. In this study, the sonographers and software performed characteristics analysis on the same group of TN ultrasound images. Then we analyzed the consistency of the two results and used the sonographer's results as the gold standard to evaluate the accuracy of SW-TH01/II.

## Materials and methods

### Participants and eligibility criteria

The research objects were thyroid ultrasound images of patients with TNs from two tertiary hospitals in Shanghai. Ultrasound images were selected by experienced sonographers from May 28 to November 22, 2021. Ultrasound examinations were performed using equipment from Philips, Siemens, Mindray, etc. The images must meet several inclusion criteria: 1) ultrasound examinations should be performed after 2017; 2) ultrasound images should be clear and complete; 3) ultrasound images must have a grey-scale cross-section without measurement markers. The images were excluded if there was unevenness in the thyroid background echo.

### Study design

This study is a pre-market registration study of the SW-TH01/II. A retrospective self-paired research was designed to verify the effectiveness and reliability of the SW-TH01/II. The SW-TH01/II and sonographers characterized TN images separately and then their results were compared for consistency. The characteristic description results of three sonographers were used as the standard to evaluate the accuracy of SW-TH01/II in analyzing the characteristics of TNs.

### Artificial intelligence software

The artificial intelligence software used in this study was developed by TenD AI Medical Technology (Shanghai) Co., Ltd. known as SW-TH01/II (version 1.0). SW-TH01/II technology is an emerging computer-aided diagnostic method. The research and development of the intelligent diagnosis system for thyroid nodule ultrasound imaging utilized a high-quality dataset comprising 5,500 ultrasound videos from 10 anatomical regions (thyroid, breast, cervical lymph nodes, axillary lymph nodes, carotid artery, liver, gallbladder, kidney, bladder, and ovary), with 550 samples per region. The dataset was randomly divided into a development set, tuning set, and test set in a 350:100:100 ratio. The image processing algorithm was the independently developed Ultrasound Super-Resolution Network (USR-Net), which incorporates modules such as skip connections, multi-scale feature fusion, and split convolution to enhance imaging processing capabilities. The localization analysis algorithm includes candidate region generation, a Multi-Scale Similarity Network (MSS-Net), and Non-Maximum Suppression (NMS). Additionally, the system employs Compute Unified Device Architecture (CUDA) and TensorRT acceleration technologies, converting core algorithms into GPU kernel functions and leveraging mixed precision computation to improve overall detection

speed. Detailed descriptions of the dataset, model architecture, training and optimization process and performance validation are provided as S1_Files. SW-TH01/II has been evaluated and validated by the results of this study, and registered with the Shanghai Drug Administration (registration number: 20212210607).

SW-TH01/II automatically detects and analyzes the characteristics of TNs based on ultrasound images. According to TI-RADS classification and ATA guideline classification, the software focused on echogenicity, shape, border, margin, and calcification of TNs. Ultrasound images are inputed into the software firstly. The sonographers then mark the approximate border box of the nodules on the image, and the software outlined the region of interest (ROI). Following the sonographer's adjustment and confirmation of the ROI, the SW-TH01/II can conduct a qualitative analysis of the characteristics of the nodules and output the results of echogenicity, shape, border, margin, and calcification. The software operation interface is shown in Fig 1.

## Sonographers evaluation

Three experienced sonographers (attending physician level and above) participated in the manual analysis of the thyroid nodule characteristics on ultrasound images, and their judgments were used as the gold standard. To ensure the reliability of the gold standard, the following procedure was implemented: Two sonographers independently provided a qualitative description of nodule characteristics from the same group of thyroid ultrasound images. If the judgments of the two sonographers were consistent, their agreement was taken as the final result. If the judgments of the two sonographers were inconsistent, a third sonographer (an associate chief physician or above) participated in the process. The third sonographer reviewed the case and facilitated a discussion to reach a consensus, which was then used as the final result. The inter-observer agreement results of the first two sonographers were reported in the S1_Table.

## Ethic

This retrospective study was approved by the ethics committees of two hospitals (approval number: 2021–038, 2021–108) and was conducted in accordance with the 1964 Helsinki Declaration and its later amendments or comparable ethical standards. Informed consent was obtained from the patients prior to obtaining ultrasound images. Subject information was obtained by identifying the outpatient or inpatient number or examination number in the hospital system. Patient-related privacy information of ultrasound images were all desensitized to protect subject information from being leaked.

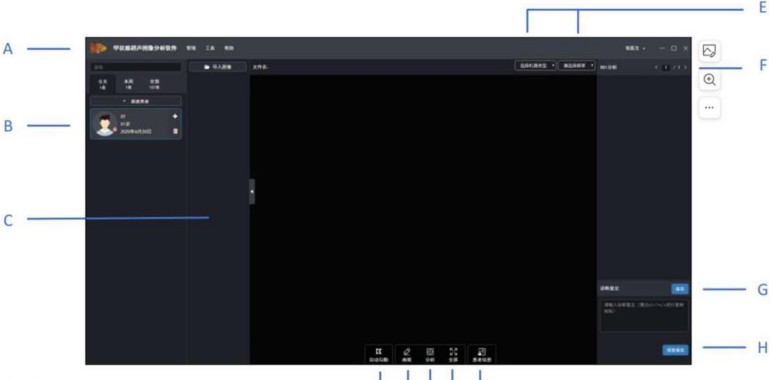

**Fig 1. Operation interface of thyroid ultrasound image analysis software.** Notes: A: menu bar (contains three main menus: management, tools, and help); B: patient list and patient retrieval; C: examination records (displays all examination records for the current patient); D: information extraction and processing (includes five functional buttons: automatic delineation, manual delineation, etc.); E: Image information (the type of machine to which the ultrasound image belongs and its frequency); F: ROI and feature extraction results; G: diagnostic opinions; H: Preview the report.

## Sample size

The expected proportion of agreement (*P*) between SW-TH01/II and the sonographer in analyzing the ultrasound image features of the thyroid gland were set at 80%. The sample size calculated by PASS 15.0 software was 407 with a confidence level (1-α) of 0.95, and an allowable error (δ) of 4%. Considering a 10% sample compensation for the quality of ultrasound images, the final total sample size was set at 450, with each medical institution having a sample size of 225 cases. The formula for sample size calculation is as follows:

$$n = \frac{Z^2_{\frac{\alpha}{2}} P(1-P)}{\delta^2}$$

n, δ, and *P* represent estimates of sample size, allowable error, and proportion of agreement, respectively.

## Statistical analyses

Categorical variables were presented by frequency (percentage), and continuous variables were described by mean, standard deviation (SD), median, percentile, minimum, maximum, and mode. The proportion of agreement and Cohen's kappa coefficient were used to evaluate the consistency between the SW-TH01/II and sonographers' analysis results. The proportion of agreement is the ratio of cases with the same judgment results as the standard judgment results to the total number of cases examined by a certain method. A proportion of agreement of at least 80% for each characteristic indicates high accuracy. The kappa coefficient is an indicator for evaluating two methods' consistency level. Kappa coefficient ≥ 0.75 indicates high consistency, and the two methods are considered equivalent; 0.4 ≤ kappa coefficient <0.75 indicates basic consistency, but further statistical analysis is required; kappa coefficient <0.4 indicates inconsistency, and the two methods are considered not equivalent [15]. The Wilcoxon signed ranks test was used to compare the analysis time of the two methods, and P < 0.05 was statistically significant. Data were analyzed using IBM SPSS Statistics ver. 26.0 (IBM Co., Armonk, NY, USA).

# Results

We planned to include 225 thyroid ultrasound images from each hospital. One image was ruled out due to failure of information desensitization. As a result, a total of 449 images were included in the final statistical analysis.

## Descriptive analysis

For the echogenicity of TNs, 116 (25.8%) and 95 (21.2%) were described by SW-TH01/II and sonographers to be homogeneous, and 333 (74.2%) and 354 (78.8%) were judged to be heterogeneous, respectively. The dominant echo of half of the images was judged to be hypoechoic level by both methods at a proportion of 50.1% and 55.9%, respectively. And there were 168 (37.4%) and 143 (31.8%) images with very hypoechoic, interpreted by SW-TH01/II and sonographers, respectively. For the aspect ratio, 121 (26.9%) and 111 (24.7%) of A/T > 1 were described by SW-TH01/II and sonographers, and 328 (73.1%) and 338 (75.3%) of A/T < 1, respectively. The SW-TH01/II interpreted 256 (57.0%) and 221 (49.2%) images with clear border and smooth margin, respectively. There were 264 (58.8%) and 224 (49.9%) cases of sonographers interpreting images with clear border and smooth margin, respectively. Both methods found that more than half of the images were calcification-free, accounting for 65.7% and 79.3%, respectively. (Table 1)

## Consistency analysis

For the seven indicators of the five characteristics of TNs (echogenicity, aspect ratio, border, margin, calcification), the proportions of agreement between SW-TH01/II and sonographers' analysis results were all greater than 80%. Taking the results of sonographer interpretation as the gold standard, the results showed that the software has a high degree of accuracy in judging the characteristics of TNs. For the echogenicity (with very hypoechoic), aspect ratio and margin, the

**Table 1. Descriptive analysis of characteristic indicators of thyroid nodules.**

| Features | SW-TH01/II | Sonographer |
|---|---|---|
|  | n (%) | n (%) |
| Echogenicity |  |  |
| Echotexture |  |  |
| Homogeneous | 116 (25.8) | 95 (21.2) |
| Heterogeneous | 333 (74.2) | 354 (78.8) |
| Echogenicity level |  |  |
| Anechoic | 62 (13.8) | 68 (15.1) |
| Hyperechoic | 15 (3.3) | 8 (1.8) |
| Isoechoic | 89 (19.8) | 101 (22.5) |
| Hypoechoic | 225 (50.1) | 251 (55.9) |
| Markedly hypoechoic | 58 (12.9) | 21 (4.7) |
| With very hypoechoic |  |  |
| Yes | 168 (37.4) | 143 (31.8) |
| No | 281 (62.6) | 306 (68.2) |
| **Aspect ratio** |  |  |
| A/T > 1 | 121 (26.9) | 111 (24.7) |
| A/T < 1 | 328 (73.1) | 338 (75.3) |
| **Border** |  |  |
| Clear | 256 (57.0) | 264 (58.8) |
| Unclear | 193 (43.0) | 185 (41.2) |
| **Margin** |  |  |
| Smooth | 221 (49.2) | 224 (49.9) |
| Unsmooth | 228 (50.8) | 225 (50.1) |
| **Calcification** |  |  |
| No calcification | 295 (65.7) | 356 (79.3) |
| Coarse calcification | 50 (11.1) | 14 (3.1) |
| Microcalcification (or combined with coarse calcification) | 104 (23.2) | 79 (17.6) |

Kappa coefficient of the software and the sonographer interpretation results were ≥ 0.75, which was statistically significant (P < 0.001), indicating high consistency. The Kappa coefficients of echogenicity (echotexture and echogenicity level), border and calcification between the two methods ranged from 0.6 to 0.75 (P < 0.001), indicating basic consistency. (Table 2) The detailed contingency tables were shown in Tables 3–9.

### Analysis time

The minimum, maximum, and mode of the software analysis time were 1s, 15s, and 3s, respectively. The median time was 3 (2, 3) seconds. The minimum, maximum, and mode of the sonographer analysis time were 13.33s, 146s, and 26s, respectively. The median time was 26.5 (21.17, 34.33) seconds. The results of Wilcoxon signed ranks test showed that the time difference between the software and sonographers to interpret the characteristic indicators of TNs was statistically significant (Z = -18.36, P < 0.001), and the analysis time of the software to interpret the characteristic indicators of TNs was lower than that of sonographers. (Table 10)

### Discussion

The main results of this study show the SW-TH01/II achieves high agreement with sonographers' analysis results, with proportions of agreement exceeding 0.8 and kappa coefficients greater than 0.6 for all features. Additionally, the software

**Table 2. Proportion of agreement and Kappa coefficient of characteristic indicators of thyroid nodules.**

| Features | Proportion of agreement | Kappa coefficient |
|---|---|---|
| Echogenicity | | |
| Echotexture | 0.895 | 0.710* |
| Echogenicity level | 0.837 | 0.749* |
| With very hypoechoic | 0.886 | 0.750* |
| **Aspect ratio** | 0.933 | 0.826* |
| **Border** | 0.849 | 0.689* |
| **Margin** | 0.878 | 0.755* |
| **Calcification** | 0.826 | 0.601* |

*P<0.001

**Table 3. Contingency Table of Agreement of Echotexture of Thyroid Nodules between SW-TH01/Ⅱ and Sonographers.**

| Echotexture | | Sonographers | | Total |
|---|---|---|---|---|
| | | Homogeneous | Heterogeneous | |
| SW-TH01/II | Homogeneous | 82 | 34 | 116 |
| | Heterogeneous | 13 | 320 | 333 |
| Total | | 95 | 354 | 449 |

Note: Data are numbers of patients.

**Table 4. Contingency Table of Agreement of Echogenicity Level of Thyroid Nodules between SW-TH01/Ⅱ and Sonographers.**

| Echogenicity level | | Sonographers | | | | | |
|---|---|---|---|---|---|---|---|
| | | Anechoic | Hyper-echoic | Isoechoic | Hypo-echoic | Markedly hypoechoic | Total |
| SW-TH01/II | Anechoic | 57 | 0 | 0 | 0 | 5 | 62 |
| | Hyperechoic | 0 | 7 | 8 | 0 | 0 | 15 |
| | Isoechoic | 0 | 1 | 85 | 3 | 0 | 89 |
| | Hypoechoic | 4 | 0 | 8 | 212 | 1 | 225 |
| | Markedly hypoechoic | 7 | 0 | 0 | 36 | 15 | 58 |
| Total | | 68 | 8 | 101 | 251 | 21 | 449 |

Note: Data are numbers of patients.

**Table 5. Contingency Table of Agreement of Very Hypoechoic of Thyroid Nodules between SW-TH01/Ⅱ and Sonographers.**

| With very hypoechoic | | Sonographers | | Total |
|---|---|---|---|---|
| | | Yes | No | |
| SW-TH01/II | Yes | 130 | 38 | 168 |
| | No | 13 | 268 | 281 |
| Total | | 143 | 306 | 449 |

Note: Data are numbers of patients.

**Table 6. Contingency Table of Agreement of Aspect Ratio of Thyroid Nodules between SW-TH01/II and Sonographers.**

| Aspect ratio | | Sonographers | | Total |
|---|---|---|---|---|
| | | A/T > 1 | A/T < 1 | |
| SW-TH01/II | A/T > 1 | 101 | 20 | 121 |
| | A/T < 1 | 10 | 318 | 328 |
| Total | | 111 | 338 | 449 |

Note: Data are numbers of patients.

**Table 7. Contingency Table of Agreement of Border of Thyroid Nodules between SW-TH01/II and Sonographers.**

| Border | | Sonographers | | Total |
|---|---|---|---|---|
| | | Clear | Unclear | |
| SW-TH01/II | Clear | 226 | 30 | 256 |
| | Unclear | 38 | 155 | 193 |
| Total | | 264 | 185 | 449 |

Note: Data are numbers of patients.

**Table 8. Contingency Table of Agreement of Margin of Thyroid Nodules between SW-TH01/II and Sonographers.**

| Margin | | Sonographers | | Total |
|---|---|---|---|---|
| | | Smooth | Unsmooth | |
| SW-TH01/II | Smooth | 195 | 26 | 221 |
| | Unsmooth | 29 | 199 | 228 |
| Total | | 224 | 225 | 449 |

Note: Data are numbers of patients.

**Table 9. Contingency Table of Agreement of Calcification of Thyroid Nodules between SW-TH01/II and Sonographers.**

| Calcification | | Sonographers | | | Total |
|---|---|---|---|---|---|
| | | No calcification | Coarse calcification | Microcalcification (or combined with coarse calcification) | |
| SW-TH01/II | No calcification | 289 | 0 | 6 | 295 |
| | Coarse calcification | 31 | 14 | 5 | 50 |
| | Microcalcification (or combined with coarse calcification) | 36 | 0 | 68 | 104 |
| Total | | 356 | 14 | 79 | 449 |

Note: Data are numbers of patients.

**Table 10. Descriptive analysis of interpretation time of thyroid nodules features.**

|  | Min | Max | Mode | 25th percentile | Median | 75th percentile | Mean | SD |
|---|---|---|---|---|---|---|---|---|
| **SW-TH01/II (s)** | 1 | 15 | 3 | 2 | 3 | 3 | 2.72 | 1.62 |
| **Sonographers (s)** | 13.33 | 146 | 26 | 21.17 | 26.5 | 34.33 | 29.71 | 12.73 |

SD: Standard deviation

significantly reduces interpretation time, requiring only about 3 seconds per image compared to 26.5 seconds for sonographers' interpretation. These findings suggest that SW-TH01/II has the potential to assist sonographers in characterizing thyroid nodule images accurately and efficiently.

Currently, ultrasound is the preliminary imaging modality in TN management [16]. Thyroid ultrasonographic characteristics can guide the initial management of TNs. Echogenicity level includes anechoic, hyperechoic, isoechoic, hypoechoic, and markedly hypoechoic, while echotexture indicates the consistency and diversity of echoes in the solid component of the nodule [1]. The shape was classified as wider-than-tall or taller-than-wide. A taller-than-wide nodule shape reflects nodule growth against normal tissue planes [17]. Irregular margins may indicate tumor infiltration of the surrounding thyroid, and extrathyroidal extension of the nodule may also be detected [18]. The probability of malignancy can be increased by all types of calcifications detected by ultrasound, as demonstrated by various research studies [19,20]. Specific ultrasound features, such as solid composition, hypo-echogenicity, irregular margins, and microcalcifications constitutes suspicious ultrasound patterns that indicate the need of prompt cytological evaluation. Thyroid ultrasonographic characteristics also determine treatment options and the type, frequency, and length of subsequent follow-up [21].

However, the sonographic manifestations of TNs are complex and diverse, and with that comes a problem: high inter- and intra-observer variability [22]. Interpretation results of ultrasound images are closely related to the knowledge and experience of sonographers [23]. Different sonographers have different understandings of TNs in the same patient, and the conclusions of the reports are very different, which brings confusion to the clinical management. Previous studies demonstrated moderate-substantial level of interobserver agreement in the evaluation of ultrasound features of TNs. Due to the difference in the allocation of medical resources, sonographers in large tertiary hospitals are faced with huge ultrasound examination tasks every day, which inevitably affects the quality of thyroid ultrasound examination, and even misdiagnosis and missed diagnosis [24].

Developments in AI technologies are taking place to overcome those limitations [25]. AI based on computer-aided diagnosis technology can improve the accuracy of diagnosis and treatment of TNs [26, 27]. Medical AI-assisted diagnosis technology is mainly used in combination with ultrasound, X-ray, CT, MR, etc., and is applied to thyroid, breast, liver, lung, muscle, carotid artery, etc [14,28,29]. Deep convolutional neural network technology is a cutting-edge AI-assisted diagnosis technology, which can automatically classify, cut and extract image features, and automatically give diagnostic results after analyzing image features [30]. Similarly, this technique is also applied in the ultrasound diagnosis of TNs [31–33]. SW-TH01/II, as a new type of AI software, uses novel machine-learning algorithms to achieve nodule characteristics extraction. After importing ultrasound images into the software, SW-TH01/II can produce standardized and objective results that help sonographers with clinical diagnosis and solve the problem of inconsistent report conclusions caused by doctors' subjective judgments. SW-TH01/II has a user-friendly interface and efficient feature recognition algorithms, allowing it to achieve faster, more accurate, and more practical results. As shown in this study, on the basis that the results of software and manual interpretation are basically consistent, the time it takes for the software to interpret an image can be shortened to about 3 seconds, which is about 27 seconds less than a manual work.

AI algorithms have demonstrated remarkable progress in image-recognition tasks motivated by the need for enhanced efficacy and performance efficiency in clinical care [34]. In the context of tertiary prevention, AI image recognition applications primarily target early detection, diagnosis, treatment response, and prognosis [35]. AI has been successfully

applied in diverse areas of medical imaging, including thoracic imaging [36], colonoscopy [37], ocular imaging (e.g., fundus photographs, diabetic retinopathy) [38], and mammography [39]. There are variations in preoperative diagnostic examinations for TNs among different levels of hospitals in China. The lack of medical resources and limited experience of ultrasound physicians in grassroots and remote areas contribute to a lower level of diagnostic standardization [40]. AI can assist in achieving more accurate diagnoses, resulting in fewer onward referrals and unnecessary fine needle aspiration [41]. AI-assisted diagnosis can undertake tedious lesion screening work, enhance accuracy, and reduce the workload on doctors. The role of AI in healthcare is defined as "augmented intelligence" by the American Medical Association, that is, AI is designed and applied to enhance human intelligence rather than replace it [42].

SW-TH01/II relies solely on static ultrasound images for feature extraction and analysis. This design has inherent limitations compared to dynamic video analysis, which allows sonographers to observe nodules in real-time and across various angles. Dynamic imaging can provide additional details, such as vascular flow patterns and the interaction of nodules with surrounding tissues, which are not captured in static images. However, the decision to evaluate the AI model on static images was based on several practical considerations. First, static images are widely used in clinical practice as a standard format for documentation and retrospective analysis. They are commonly archived and reviewed for diagnostic purposes, making static image analysis highly relevant to real-world workflows. Second, static images offer greater standardization and reproducibility, minimizing variability caused by operator-dependent factors during image acquisition. This is particularly important for multi-center studies, where standardization is critical for ensuring consistent and comparable results across different datasets. Third, static image analysis simplifies the computational and data storage requirements, which can be a significant challenge when working with large volumes of dynamic video data. By focusing on static images, we were able to streamline data collection and analysis while maintaining consistency across the study. Despite these advantages, we acknowledge the limitations of relying solely on static images. Dynamic imaging undoubtedly provides richer diagnostic information, and future iterations of SW-TH01/II could incorporate dynamic video analysis to further enhance its diagnostic capabilities.

The heterogeneity of ultrasound images, arising from variations in machine models, imaging settings, and patient-specific factors, poses a significant challenge to the performance and generalizability of AI algorithms. Differences in image quality, resolution, frequency ranges, and gain settings can lead to inconsistencies in feature extraction and anomaly detection. For example, linear probes with high frequencies generate detailed images suitable for shallow regions like the thyroid, while convex probes with lower frequencies produce deeper but less detailed images, often resulting in variations in texture and contrast. To address this, our study incorporated a diverse dataset, ensuring representation from ten different machine brands (e.g., GE, Siemens, Mindray, etc.) and various imaging conditions (e.g., frequency ranges, gain settings). Additionally, preprocessing with the Ultrasound Super-Resolution Network (USR-Net) was employed to standardize image quality, reduce noise, and minimize the influence of heterogeneity. While the preprocessing effectively harmonized images across different sources, residual variations may still impact the algorithm's performance in subtle ways. Future efforts should focus on evaluating model performance across specific dimensions of heterogeneity, such as machine-specific variations or extreme imaging conditions, to further enhance robustness and clinical applicability.

One of the key advantages of SW-TH01/II is its ability to analyze thyroid ultrasound images significantly faster than manual interpretation by sonographers. Its rapid analysis may raise questions about the trade-off between speed and diagnostic accuracy. It is important to emphasize that the primary role of SW-TH01/II is to serve as an auxiliary tool, assisting rather than replacing sonographers in diagnostic workflows. AI-based SW-TH01/II is designed to enhance standardization and reproducibility in feature analysis, reducing variability caused by human factors such as subjective judgment or fatigue. By automating the extraction and analysis of key image features, the software aims to reduce the workload of clinicians and improve overall efficiency, allowing them to focus on cases that require more detailed interpretation. Therefore, the software's speed is not a substitute for accuracy but rather a means to streamline routine tasks and support clinicians in making more informed and timely decisions. Nevertheless, future studies should further evaluate the software's performance in real-world clinical settings to confirm this balance and identify potential areas for optimization.

The study has some limitations. First, the images included in this study were derived from retrospective samples of clinical centers, which introduces some selection bias. However, this did not affect the comparison of the identified results between the software and the sonographers. Second, the study exclusively focused on the patients' TN images and did not collect their basic demographic or clinical information, leading to the absence of participant-related details in the results section. These details may influence nodule characteristics and diagnostic outcomes, and their inclusion could provide a more comprehensive evaluation of the software's performance. Future studies should aim to incorporate such data to better understand how patient-specific variables impact both AI and sonographer interpretations.

Meanwhile, the software has its limitations. On the one hand, SW-TH01/II, as a Class II medical device, is only able to analyze the characteristics of TN. It is not fully indicative of either benign or malignant nodules. On the other hand, sonographers could improve performance by reading dynamic videos instead of static images only, but the software can only analyze static images. So in a real-world setting, the final diagnosis should still be made by sonographers. SW-TH01/II can only play an assisting role. Therefore, the cooperation between sonographers and software to provide the final diagnosis is more suitable for the clinical setting. The application scenario of SW-TH01/II in the real world should be community healthcare centers, rather than large general hospitals to help train clinical doctors and improve the standard of medical care in remote areas and primary healthcare facilities [43]. The clinical adoption of AI-based diagnostic tools like SW-TH01/II faces several challenges. First, the integration of AI into clinical workflows requires extensive training for healthcare providers to interpret AI outputs effectively and to address potential discrepancies between AI and human diagnoses. Second, ethical concerns surrounding AI deployment must be carefully considered, particularly regarding data privacy, algorithm transparency, and accountability in cases of misdiagnosis. Additionally, regulatory approval processes and cost considerations may limit the widespread use of such technologies, especially in resource-limited settings. To address these challenges, future improvements should focus on enhancing the explainability of AI models, ensuring compliance with ethical and legal standards, and reducing the cost of implementation to increase accessibility.

## Conclusions

SW-TH01/II, an AI-assisted ultrasound tool designed for analyzing TN features, can offer objective and standardized results. It serves as a valuable diagnostic reference for sonographers, enhancing their diagnostic accuracy. This tool is expected to improve ultrasound diagnosis in primary healthcare, enhance examination efficiency, and reduce the workload for ultrasound workers. Despite these promising findings, further prospective clinical validation is essential for confirming the software's clinical applicability and ensuring its broader adoption in routine practice. With its broad clinical application prospects and further research value, AI-assisted software holds significant potential.

## Supporting information

**S1 Files. The research and development of the intelligent diagnosis system for thyroid nodule ultrasound imaging.** This supplementary material provides detailed information on the methods, algorithms, and system architecture used in the development of the intelligent diagnosis system for thyroid nodule ultrasound imaging.
(PDF)

**S1 _Table. Proportion of agreement of characteristic indicators between two sonographers.** This table presents the proportion of agreement between two sonographers in evaluating characteristic indicators of thyroid nodules based on ultrasound imaging.
(DOCX)

**S1 _Dataset. Dataset for Statistical Analysis of Thyroid Nodule Characteristics.**
(XLSV)

## Acknowledgments

We are grateful to all the company's R&D personnel for their extensive work in software development, and the two medical institutions for providing valuable ultrasound image data and strong support for the research.

## Author contributions

**Conceptualization:** Ying Wang, Yong Cai.

**Data curation:** Zhongqing Xu.

**Formal analysis:** Chen Xu, Zuxin Wang, Fan Hu.

**Funding acquisition:** Ying Wang, Zhongqing Xu, Yong Cai.

**Investigation:** Jun Zhou.

**Methodology:** Chen Xu, Yong Cai.

**Project administration:** Jun Zhou.

**Resources:** Jun Zhou, Fan Hu, Ying Wang, Zhongqing Xu, Yong Cai.

**Software:** Jun Zhou.

**Supervision:** Fan Hu, Ying Wang, Yong Cai.

**Writing – original draft:** Chen Xu, Zuxin Wang.

**Writing – review & editing:** Chen Xu, Zuxin Wang, Fan Hu, Ying Wang, Zhongqing Xu.

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
