## [Decision Letter · Decision Letter 0]

19 Feb 2025

PONE-D-25-01972
Application research of artificial intelligence software in the analysis of thyroid nodule ultrasound image characteristics
PLOS ONE

Dear Dr. Cai,

Thank you for submitting your manuscript to PLOS ONE. After careful consideration, we feel that it has merit but does not fully meet PLOS ONE’s publication criteria as it currently stands. Therefore, we invite you to submit a revised version of the manuscript that addresses the points raised during the review process.

We look forward to receiving your revised manuscript.

Kind regards,

Fahad Farhan Almutairi, PhD

Academic Editor

PLOS ONE

Journal Requirements:

“This study was supported by the Clinical Research Project of Shanghai Municipal Health Commission (202240198), Key Discipline Projects of Shanghai Three-Year Action Plan for Public Health under Grant (GWVI-11.1-29), Science and Technology Commission Shanghai Municipality (Grant 20JC1410204) for the Seroepidemiological Study of Novel Coronavirus Pneumonia in Key Populations, Key Supporting Disciplines of Shanghai Health System (Grant Number-2023ZDFC0403), Shanghai Health Care Commission Clinical Research Program (20214Y0205).”

Reviewers' comments:

Reviewer's Responses to Questions

**Comments to the Author**

1. Is the manuscript technically sound, and do the data support the conclusions?

Reviewer #1: Yes

Reviewer #2: Yes

2. Has the statistical analysis been performed appropriately and rigorously? 

Reviewer #1: Yes

Reviewer #2: Yes

3. Have the authors made all data underlying the findings in their manuscript fully available?

Reviewer #1: Yes

Reviewer #2: No

4. Is the manuscript presented in an intelligible fashion and written in standard English?

Reviewer #1: Yes

Reviewer #2: Yes

5. Review Comments to the Author

Reviewer #1: The manuscript demonstrates significant scientific merit, and its findings contribute to the growing application of AI in thyroid imaging. Addressing the above points will enhance the manuscript's clarity, completeness, and credibility.

Reviewer #2: 1- Data Availability: Ensure compliance with PLOS ONE’s open data policy by making an anonymized dataset available.

2- Static vs. Dynamic Images: Justify why the AI model is evaluated only on static images.

3- Patient Demographics: Include age, gender, and clinical factors in the dataset description.

4- Subgroup Analysis: Analyze AI performance across different image sources and machines.

5- Gold Standard Reliability: Report inter-observer agreement among sonographers.

6- AI Model Training Details: Clarify training dataset size, architecture, and validation method.

7- Time vs. Accuracy Trade-off: Discuss whether AI’s faster speed affects its diagnostic quality.

8- Expand Limitations Section: Discuss clinical adoption challenges, ethical concerns, and future improvements.

6. PLOS authors have the option to publish the peer review history of their article (what does this mean?). If published, this will include your full peer review and any attached files.

Reviewer #1: **Yes: **Luqman Adewale Abass

Reviewer #2: **Yes: **AMJED ABBAS

---

## [Author Response · Author response to Decision Letter 0]

4 Apr 2025

Dear Editor and Reviewers,

We sincerely appreciate your thoughtful comments and constructive suggestions on our manuscript (PONE-D-25-01972). Your detailed feedback has been invaluable in improving the quality and clarity of our work. We have carefully addressed each of your comments and revised the manuscript accordingly.

In this rebuttal letter, we have provided detailed replies to all the points raised, with our responses formatted in steel blue for clarity. Additionally, we have highlighted the major changes in the revised manuscript using lapis blue to ensure they are easily identifiable.

Thank you again for your time and effort in reviewing our paper. We hope that the revisions meet your expectations and address your concerns.

Sincerely,

Yong Cai

Point-to-point responses are made as follows:

Comments from Academic Editor

1. Please ensure that your manuscript meets PLOS ONE’s style requirements, including those for file naming.

Response:

Thank you for your feedback regarding the manuscript’s compliance with PLOS ONE’s style requirements. We have carefully reviewed the journal’s style guidelines and have adjusted the manuscript’s structure accordingly to ensure full compliance. If there are any additional areas that require further adjustments, please let us know, and we will address them promptly.

2.Please note that PLOS ONE has specific guidelines on code sharing for submissions in which author-generated code underpins the findings in the manuscript. In these cases, all author-generated code must be made available without restrictions upon publication of the work.

Response:

Thank you for pointing out PLOS ONE’s guidelines on code sharing. We confirm that no author-generated code was used in this study, as the findings are based entirely on the analysis of collected data and the application of existing statistical methods. Therefore, there is no code to share for this manuscript. If further clarification is needed, please let us know.

3.Please state what role the funders took in the study. If the funders had no role, please state: “The funders had no role in study design, data collection and analysis, decision to publish, or preparation of the manuscript.” If this statement is not correct you must amend it as needed. Please include this amended Role of Funder statement in your cover letter; we will change the online submission form on your behalf.

Response:

Thank you for your feedback regarding the Role of Funder statement. We have revised the financial disclosure statement in the manuscript to include a description of the funders’ contributions. Specifically, we have added the following sentence: “The funders contributed to the study design, data collection and analysis, preparation of the manuscript, and the decision to publish.”Additionally, we have included this information in the cover letter as part of the financial disclosure statement. This addition ensures that the role of the funders is clearly and transparently stated in accordance with PLOS ONE’s guidelines. Please let us know if further modifications are required.

4.We note that you have indicated that there are restrictions to data sharing for this study. For studies involving human research participant data or other sensitive data, we encourage authors to share de-identified or anonymized data. However, when data cannot be publicly shared for ethical reasons, we allow authors to make their data sets available upon request. Please update your Data Availability statement in the submission form accordingly.

Response:

Thank you for your feedback regarding the Data Availability Statement. We have updated the statement in the manuscript to reflect that the data set used for analysis in this study has been uploaded as Supporting Information files. The revised statement now reads as follows: “The data set used for analysis in this study has been uploaded as Supporting Information files.” This ensures compliance with PLOS ONE’s data sharing policies. Please let us know if further adjustments are required.

5.Please review your reference list to ensure that it is complete and correct. If you have cited papers that have been retracted, please include the rationale for doing so in the manuscript text, or remove these references and replace them with relevant current references. Any changes to the reference list should be mentioned in the rebuttal letter that accompanies your revised manuscript. If you need to cite a retracted article, indicate the article’s retracted status in the References list and also include a citation and full reference for the retraction notice.

Response:

Thank you for your feedback regarding the reference list. We have carefully reviewed all references cited in the manuscript to ensure that they are complete, correct, and up to date. We confirm that none of the cited papers have been retracted. If any retracted articles were to be identified in future revisions, we will make the necessary updates to the reference list and provide a rationale for their inclusion or replace them with relevant current references, as per the journal’s guidelines. Please let us know if there are any additional concerns regarding the references.

Comments from Reviewer 1

The manuscript demonstrates significant scientific merit, and its findings contribute to the growing application of AI in thyroid imaging. Addressing the above points will enhance the manuscript's clarity, completeness, and credibility.

1.Provide additional information on the training dataset size, algorithm optimization, and performance validation metrics.

Response:

Thank you for your valuable comments regarding the AI model training details. We have provided a detailed explanation addressing the training dataset size, model architecture, and validation methods below:

(1)Dataset Details:

The intelligent diagnosis system for thyroid nodule ultrasound images was developed using a high-quality and diverse dataset consisting of 5,500 ultrasound video samples from 10 anatomical regions (thyroid, breast, cervical lymph nodes, axillary lymph nodes, carotid artery, liver, gallbladder, kidney, bladder, and ovary). Each region contributed 550 video samples, which were randomly divided into a development set (350 samples), a tuning set (100 samples), and a test set (100 samples) in a 350:100:100 ratio.

Specifically, for thyroid-related data, 550 ultrasound videos were divided into a development set (350 samples), a tuning set (100 samples), and a test set (100 samples). The training and validation processes utilized these subsets to ensure a robust evaluation of the AI model.

(2)Image Processing Algorithm:

The image processing algorithm in this system employed the Ultrasound Super-Resolution Network (USR-Net). This network architecture comprises:147 layers with approximately 5 million parameters. Key features include skip connections and multi-scale feature fusion to enhance high-dimensional feature extraction efficiency. Split convolution was used to improve the ability to extract fine details and textures, while channel fusion was integrated to optimize the processing speed for dynamic ultrasound images. The image processing algorithm was trained using the development set, with low-quality images generated through degradation processing as inputs and the corresponding high-quality original images as targets. The optimization objective was to minimize the mean absolute error (MAE). After 500 epochs of training, the MAE stabilized, achieving convergence.

(3)Localization Analysis Algorithm:

The localization analysis algorithm consisted of three components: Candidate Region Generation, Multi-Scale Similarity Network (MSS-Net) and Non-Maximum Suppression (NMS). Key characteristics of the MSS-Net included 130 layers with approximately 4 million parameters, utilizing depthwise separable convolution to enhance computational efficiency. The NMS process employed an IOU threshold of 0.7 to select the final localization results, which significantly improved precision.

(4)Validation and Optimization:

The dataset was divided into a development set for training and a tuning set for parameter optimization. Performance validation was conducted using the independent test set. The degradation process for training involved generating low-quality images as input and using the original high-quality images as the training target. The KMeans clustering algorithm was used in the candidate region generation phase to select the optimal window size (K=3). During the multi-scale similarity detection phase, the sigmoid loss function was optimized. The IOU threshold for NMS was determined based on ROC curve analysis, with 0.7 being the optimal value. To meet real-time dynamic analysis requirements, Compute Unified Device Architecture (CUDA) and TensorRT acceleration technologies were introduced, converting core algorithms into GPU kernel functions and leveraging mixed precision computation. This improved the processing speed by a factor of two.

(5)Performance Results:

The image processing algorithm improved the Peak Signal-to-Noise Ratio (PSNR) from 21.16 (before processing) to 35.15 (after processing) and the Structural Similarity Index (SSIM) from 0.582 to 0.967. Statistical analysis (t-test) showed no significant differences in image quality across ultrasound machine brands, frequency ranges, and gain ranges after processing (p > 0.05), demonstrating effective image quality normalization. The localization analysis algorithm achieved a sensitivity of 95.04%, specificity of 94.17%, and accuracy of 94.74% on the test set. For thyroid-specific data, the sensitivity was 94.14%, specificity was 95.45%, and accuracy was 94.63%. The system’s average processing time per frame was 18.72 milliseconds (maximum 25 milliseconds), meeting the real-time processing requirement of completing each frame analysis within 33 milliseconds.

We hope this detailed explanation addresses your concerns. Due to space limitations, we have provided detailed descriptions of the dataset, model architecture, training and optimization process and performance validation as Supporting Information 1. In the revised manuscript, we have summarized these key points in the Materials and methods section.

2.Clarify the potential impact of image heterogeneity on AI performance.

Response:

Thank you for your insightful comment regarding the potential impact of image heterogeneity on the performance of the AI algorithm. We fully acknowledge that variations in image characteristics—caused by factors such as different machine models, imaging settings, and patient-specific conditions—can influence the performance of AI algorithms. In our study, we specifically addressed this challenge by incorporating diverse data sources during the algorithm development phase. The dataset included ultrasound images collected from ten different machine models (e.g., GE, Siemens, Mindray, Esaote, Hitachi, Toshiba, Philips, Samsung, Neusoft, VINNO) with varying frequency ranges and gain settings. These variations ensured the representation of a wide spectrum of imaging conditions, contributing to the algorithm's robustness and generalizability.

To mitigate the impact of heterogeneity, we utilized our proprietary Ultrasound Super-Resolution Network (USR-Net) to preprocess the images. This network effectively reduced noise, standardized image quality, and harmonized variations caused by different machine brands, frequency ranges, and gain settings. Postprocessing analysis, including PSNR and SSIM evaluations, showed that the processed images achieved consistent quality across different imaging conditions, with no statistically significant differences (p > 0.05). This preprocessing step minimized the influence of image heterogeneity on downstream tasks, such as anomaly detection and localization.

While our current approach successfully reduced the impact of image heterogeneity, we recognize that residual variations may still exist and could affect the AI's performance in subtle ways. Future studies will focus on more granular analyses of algorithm performance across specific sources of heterogeneity, including individual machine models and imaging settings, to further optimize the algorithm's adaptability to diverse clinical scenarios. We have expanded the discussion section of the manuscript to include a detailed analysis of the potential impact of image heterogeneity on AI performance.

3.Reframe the conclusion to acknowledge the need for prospective clinical validation.

Response:

We sincerely thank for the valuable feedback and suggestions. Based on your recommendation, we have revised the conclusion to acknowledge the need for further prospective clinical validation of the SW-TH01/II AI-assisted ultrasound tool. In the revised conclusion, we emphasize the importance of further prospective clinical validation “ Despite these promising findings, further prospective clinical validation is essential for confirming the software’s clinical applicability and ensuring its broader adoption in routine practice”.

4.Discuss potential limitations of the software's static image-based analysis.

Response:

Thank you for raising the important point regarding the limitations of relying on static images for analysis. We agree that static image-based analysis has inherent constraints compared to dynamic video analysis. In the revised manuscript, we have acknowledged this limitation and provided a justification for the use of static images in our study. Specifically, we stated: “SW-TH01/II relies solely on static ultrasound images for feature extraction and analysis. This design has inherent limitations compared to dynamic video analysis, which allows sonographers to observe nodules in real-time and across various angles. Dynamic imaging can provide additional details, such as vascular flow patterns and the interaction of nodules with surrounding tissues, which are not captured in static images. However, the decision to evaluate the AI model on static images was based on several practical considerations. First, static images are widely used in clinical practice as a standard format for documentation and retrospective analysis. They are commonly archived and reviewed for diagnostic purposes, making static image analysis highly relevant to real-world workflows. Second, static images offer greater standardization and reproducibility, minimizing variability caused by operator-dependent factors during image acquisition. This is particularly important for multi-center studies, where standardization is critical for ensuring consistent and comparable results across different datasets. Third, static image analysis simplifies the computational and data storage requirements, which can be a significant challenge when working with large volumes of dynamic video data. By focusing on static images, we were able to streamline data collection and analysis while maintaining consistency across the study. Despite these advantages, we acknowledge the limitations of relying solely on static images. Dynamic imaging undoubtedly provides richer diagnostic information, and future iterations of SW-TH01/II could incorporate dynamic video analysis to further enhance its diagnostic capabilities.”

5.Conduct a thorough language review to correct minor errors.

Response:

Thank you for pointing out the need for a thorough language review. We have carefully reviewed the entire manuscript to identify and correct minor language errors, improve clarity, and ensure consistency in terminology. We also paid special attention to grammatical accuracy, sentence structure, and overall readability.

6.Revise technical terms for consistency (e.g., “dectects” should be “detects”).

Response:

Thank you for noticing the inconsistency in technical terms and pointing out the specific example. In response to your comment, we have thoroughly reviewed the manuscript to ensure that all technical terms are used consistently and correctly throughout the text. Specifically, the typo “dectects” has been corrected to “detects,” along with a careful examination of other similar terms to avoid any remaining errors. We appreciate your attention to detail, which has helped us improve the overall quality and precision of the manuscript.

7.Th

---

## [Editor Report · Decision Letter 1]

8 Apr 2025

Application research of artificial intelligence software in the analysis of thyroid nodule ultrasound image characteristics

PONE-D-25-01972R1

Dear Dr. Cai,

We’re pleased to inform you that your manuscript has been judged scientifically suitable for publication and will be formally accepted for publication once it meets all outstanding technical requirements.

Kind regards,

Fahad Farhan Almutairi, PhD

Academic Editor

PLOS ONE
---

## [Editor Report · Acceptance letter]

PONE-D-25-01972R1

PLOS ONE

Dear Dr. Cai,

I'm pleased to inform you that your manuscript has been deemed suitable for publication in PLOS ONE. Congratulations! Your manuscript is now being handed over to our production team.

Kind regards,

on behalf of

Dr. Fahad Farhan Almutairi

Academic Editor

PLOS ONE